



# Comprehensive Factors Influencing Lateral Soil Water Flow Patterns on Hillslopes: Insights from Experimental and Simulation Studies

Wande Gao[1,2,3], Yandong Ma[4], Xiuhua Liu[1,2,3], Yudong Lu[1,2,3], Ce Zheng[1,2,3], Yunfei Chen[1,2,3], Cunping Ma[5]

[1]Key Laboratory of Subsurface Hydrology and Ecological Effect in Arid Region of Ministry of Education, Chang'an University, Xi'an 710054, China
[2]School of Water and Environment, Chang'an University, Xi'an 710054, China
[3]Key Laboratory of Eco-hydrology and Water Security in Arid and Semi-arid Regions of the Ministry of Water Resources, Chang'an University, Xi'an 710054, China
[4] Key Laboratory of State Forestry Administration on Soil and Water Conservation & Ecological Restoration of Loess Plateau, Shaanxi Academy of Forestry, Xi'an 710082, China
[5]China State Long-Term Observation and Research Station for Mu Us Desert Ecosystem in Yulin of Shaanxi, Yulin 719000, China

*Correspondence to*: Xiuhua Liu (liuxh68@chd.edu.cn) and Ce Zheng (Zhengce@chd.edu.cn)

**Abstract.** This study conducted a rainfall tracer experiment on the slope to investigate water flow patterns. Three distinct water flow patterns were observed within the slope, including lateral upslope and vertical flow during rainfall, followed by lateral downslope flow after rainfall. A series of scenario simulations were conducted to enhance understanding of the comprehensive factors influencing lateral water flow on the slope. These simulations considered various factors, including two rainfall patterns (RP), three soil types (ST), three slope angles (SA), three anisotropy ratios (AR), and two layered slope systems (LS). The Hydrus-2D model was applied based on the two-dimensional Richards equation under given initial soil water and boundary conditions. Spatiotemporal variations in horizontal flux ($q_x$), vertical flux ($q_z$), and deviation of the water flow direction from vertical (DWFFV) in the central slope profile were calculated. The simulations of water flow on slopes under different conditions also confirmed three distinct water flow patterns manifesting within slopes during rainfall. The results indicated complete consistency between the direction of lateral water flow (lateral unsaturated upslope or downslope flow) and soil water potential horizontal gradients ($\partial\varphi/\partial x$). This suggests that the direction of lateral water flow is regulated by $\partial\varphi/\partial x$ rather than the change in water content over time ($\partial\theta/\partial t$). The rate of movement of the wetting front differed among soil types, with the highest in sand. DWFFV and the corresponding $q_x$ value showed positive correlations with SA and AR.

## 1 Introduction

Lateral water flow in the unsaturated zone has an important effect on the redistribution of soil water in the deep groundwater area. The lateral redistribution of soil water and surface convergence along the topographic slope can result in soil water accumulation at a depth of 1 to 2 m at the foot of slopes, which is critical for biological, geomorphological, and chemical





processes (Morbidelli et al., 2012; Kidron, 1999). Lateral water flow affects the spatial distributions of vegetation in arid areas (Gao et al., 2022), alters the transport paths of subsurface contaminants (Flury, 1996; Ju and Kung, 1997), and also

plays an important role in compromising slope stability, potentially leading to landslide or surface erosion (Wienhöfer et al., 2011; Loáiciga and Johnson, 2018).

Soil water distribution in hillslope is a transient, variably saturated, physical process regulated by rainfall characteristics, slope geometry, and hydrological properties of slope media (Lu et al., 2011; Wang and Chen, 2021). The significant changes in rainfall intensity during complex rainfall may lead to multiple processes acting to redistribute soil water (Corradini et al.,

1994, 1997), resulting in complex slope water flow patterns (Novakowski and Gillham, 1988; Wu et al., 2021). For a rainfall event of an intensity that exceeds saturated hydraulic conductivity, given the differences in infiltration capacity and permeability of soil types, the proportions of rainfall allocated to surface runoff and soil water are dependent on soil type. The migration rates of wetting front on slopes also vary among different soil types and shows positive relationships with soil particle size. Slope angle directly influences the amount of rainfall infiltrating the surface (Morbidelli et al., 2015), the water

head at the wetting front, and the direction of infiltration of soil water on the slope (Morbidelli et al., 2016; Wu et al., 2017). Under the assumption of vertically falling rainfall and among slopes with the same slope length, steep slopes receive less rain, which may directly affect infiltration and runoff generation (Chen and Young, 2006). The head at the wetting front becomes smaller as the slope angle increases, and infiltration capacity on steep slopes tends to decrease (Wu et al., 2018). And the more steepness the slope, the greater the angle between the matric potential gradient and the gravitational potential

gradient at the wetting front, which changes the resultant force of gravitational and matric potential, thus the direction of water movement (Lu et al., 2011). Soil layering and anisotropy are additional essential factors promoting lateral flow (Mccord and Stephens, 1987; Warrick et al., 1997).

A research frontier within hydrological studies has been the quantitative expression of the process of lateral soil water flow on slope. While the semi-analytical representation for distributions of transient water of slopes with constant initial

water content proposed by Philip (1991) are not applicable post rainfall, it remains of relevance for demonstrating that a slope with homogeneous and isotropic soils cannot experience true lateral downslope water flow during wetting. The observation of such flow in the slope field indicates that either the flow does not conform to the Richards equation or the presence of changes due to stratification or anisotropy. Jackson (1992) used numerical methods to solve the Richards equation and simulated rainfall infiltration on a homogeneous and isotropic slope. The results of the Jackson (1992) study

showed that the wetting front initially moves toward the direction normal to the slope during the initial rainfall stage. After the onset of rainfall, water near the slope surface gradually changes direction, moving vertically downwards. Water flow behind the wetting front will continue to move toward the direction normal to the slope. After the cessation of rainfall, the slope surface becomes a no-flow boundary (ignoring evaporation), and the shallow unsaturated flow moves almost parallel to the slope surface. Sinai and Dirksen (2006) verified the occurrence of unsaturated lateral flow under rainfall using a

laboratory sandbox unsaturated infiltration test. Their findings indicated that rainfall cessation was not a requirement for the generation of unsaturated lateral flow in the downslope direction, consistent with those obtained by Zaslavsky and Sinai





(1981) Lu et al. (2011) proposed a mechanism to determine the slope water flow direction in a homogeneous and isotropic slope based on three slope water states: (1) a drying state ($\partial\theta/\partial t < 0$) during which lateral downslope flow occurs; (2) a steady state ($\partial\theta/\partial t = 0$) during which water flows vertically downward; (3) a wetting state ($\partial\theta/\partial t > 0$) during which

lateral upslope flow occurs. However, at a given time, the direction of water flow at a point within a slope may be independent of the change in water content and may in fact depend on the spatial distribution of soil water potential in the slope.

   Although previous studies have suggested that prevailing rainfall conditions could completely regulate the direction of flow along the lateral downslope, lateral upslope, or vertically downward (Jackson, 1992; Sinai and Dirksen, 2006; Lu et al.,

2011; Lv et al., 2013), the conditions required for lateral downslope unsaturated flow in hillslopes remain poorly understood. Observations of soil water movement on a slope differ due to differences in topographic factors and precipitation distributions. Under time-varying rainfall conditions, concurrent vertical, downslope, and upslope lateral flow can occur at different depths and locations within the hillslope. In addition, the direction of soil water flow may be regulated by trade-off between gravity potential and matrix potential. The objectives of the present study were to: (1) identify a general mechanism

regulating the direction of rainfall-induced lateral slope water flow; (2) investigate the effect of different rainfall patterns (RP), slope angles (SA), soil types (ST), anisotropy ratios (AR), and layered slope systems (LS) on slope water flow. The result of the present study can reveal the process of water movement in slopes under complex conditions and the various influencing factors, thereby enhancing understanding of slope hydrological processes.

## 2 Methodology

### 2.1 Rainfall dye tracer experiments

   The present study conducted dye tracer (brilliant blue) experiments on the dune slopes (average gradient of 29°) in Mu Us Sandy Land, China (for details location see (Gao et al., 2022)) to investigate the slope soil water flow pattern. The experiment was conducted on a plot size of 20 cm × 40 cm, with three points selected along the slope from right to left, designated Point 1, Point 2, and Point 3 (Fig. 1a). A medical intravenous syringe was used to inject 1-g L$^{-1}$ brilliant blue dye

solution onto the slope surface at a controlled rate of 30 drops per minute. This allowed for qualitative assessment of the soil water flow process based on the stained area profile. Simultaneously, a hand-operated pressurized spray bottle was used to simulate rainfall, with 2 L of water sprayed within 8 min, equivalent to a total rainfall of 2.5 cm, at an average rate of 0.3125 cm min$^{-1}$. At the end of the simulated rainfall, the soil water flow pattern on the slope was observed by immediately excavating Point 1. The stained area was subsequently photographed using a digital camera. A similar procedure was

repeated for excavation at Point 2 and Point 3, with these excavations being conducted one hour and two hours after the simulated rainfall had ended, respectively. A total of three sets of experiments were conducted, and the results of two of these experiment sets were illustrated in Fig. 1c–e and Fig. 1f–h, respectively.





Moreover, to visually observe the lateral movement of soil water on the slope under multiple natural rainfall events, we conducted a dye tracer experiment on the same sand dune slope on May 15, 2022. Dye was horizontally embedded at depths

of 2 cm, 5 cm, 10 cm, 20 cm, and 50 cm at the top, middle, and bottom positions of the slope, with a length of 10 cm for each depth, as shown in Fig. 2a and Fig. 2b.

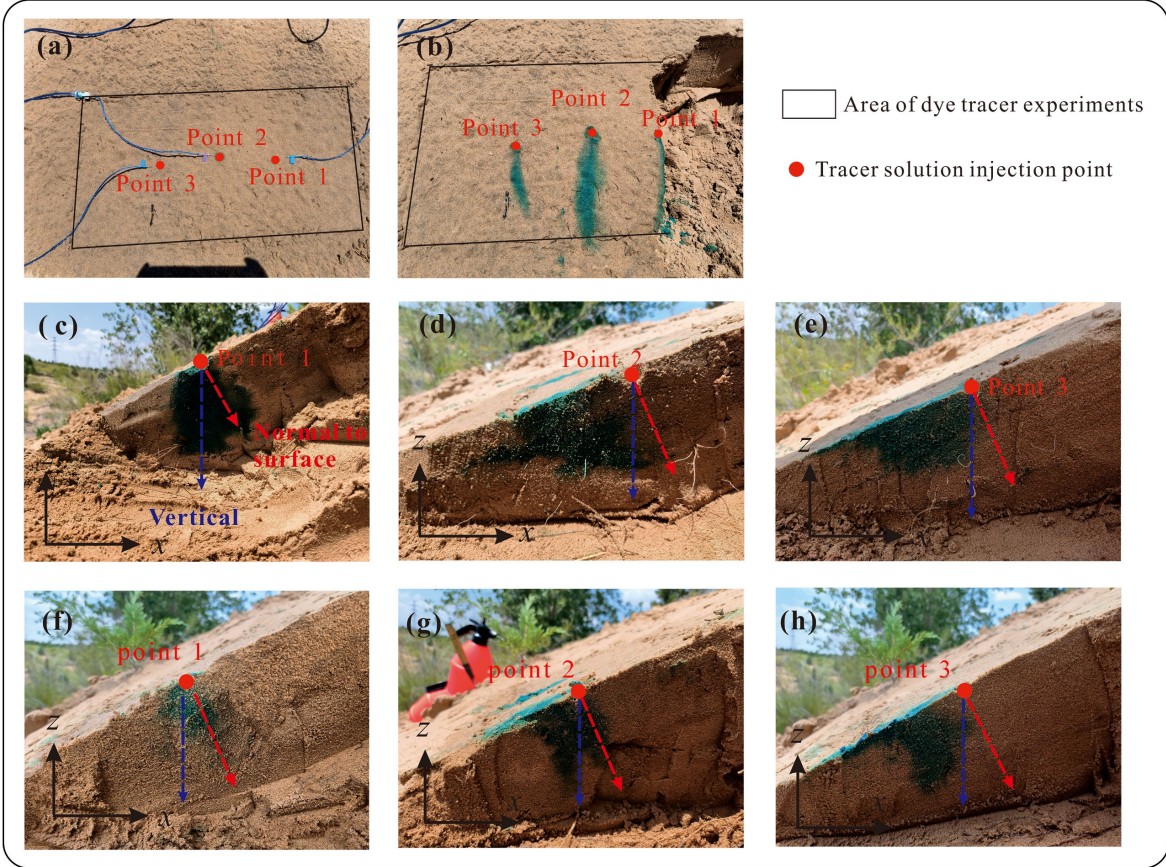

**Figure 1.** Visualization of the flow path for simulated rainfall dye tracer experiments on slopes. (a) layout of the simulated rainfall dye tracer experiments at the end of a simulated rainfall event; (b) stained traces on the slope surface 40 min after

rainfall (first experiment set); (c), (f) visualization of the flow path at Point 1 at the end of the rainfall tracer experiment; (d), (g) visualization of the flow path at Point 2 one hour after the end of the rainfall tracer experiment; (e), (h) visualization of the flow path at Point 3 two hours after the end of the rainfall tracer experiment. (c) – (f) represent the results of the first experiment set; (g) – (h) represent the results of the second experiment set.





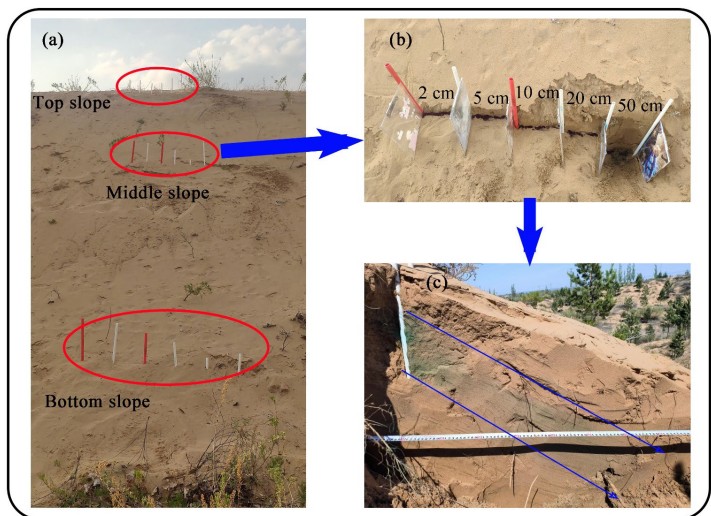

**Figure 2.** (a) Layout of tracer experiments under natural rainfall on the slope. (b) Layout of dye embedded at middle slope. (c) Visualization of flow path at 50 cm.

## 2.2 Simulation design

A series of scenario simulations were conducted to comprehensively analyze the factors influencing lateral flow and water infiltration on slopes. These simulations were designed to provide a deeper understanding of the dynamics governing lateral water movement and the complex interactions between multiple factors affecting water infiltration on sloped terrains. The scenarios investigated were two RP, three ST, three SA, three AR, and two LS control methods.

The two RP investigated were: (1) rainfall intensity in the first 8 h equal to 0.5 cm h$^{-1}$ (constant rainfall intensity); (2) rainfall intensity varying between 0.4 cm h$^{-1}$ and 0.6 cm h$^{-1}$ in an 8-hour period, changing on an hourly basis while remaining constant within each hour (variable rainfall intensity). Both RP had a total depth of 4 cm, and the simulation extended from the start of rainfall to 4 h after cessation of rainfall. No runoff from the slope occurred during the rainfall process. The ST selected in the present study were sand (Soil 1), sandy loam (Soil 2), and silt loam (Soil 3). The present study utilized sand (soil 1) with the same characteristics as sand dunes in which the dye tracer experiments were conducted. The soil hydraulic parameters for soil 1 were chosen according to previous studies conducted in the same study area (Zheng et al., 2020). Additionally, sandy loam (Soil 2) (Lv et al., 2013) and silt loam (Soil 3) (Liu et al., 2018) were selected as contrasting soil types to enable a comprehensive comparison. Table 1 presents hydraulic parameters for soil used in the simulation design. Table 2 provides the values employed in designing the scenarios. Additionally, two layered slope models (Soil 1/Soil 3 and Soil 3/Soil 1) were established to examine lateral flow in layered slopes.

**Table 1.** Hydraulic parameters for different soil types in the simulation design.





|  | $\theta_r$ (cm³ cm⁻³) | $\theta_s$ (cm³ cm⁻³) | $\alpha$ (cm⁻¹) | $n$ (-) | $K_s$ (cm min⁻¹) |
|---|---|---|---|---|---|
| Soil 1 (Zheng et al., 2020) | 0.011 | 0.4 | 0.028 | 1.57 | 0.783 |
| Soil 2 (Lv et al., 2013) | 0.0432 | 0.42 | 0.025 | 1.9 | 0.155 |
| Soil 3 (Liu et al., 2018) | 0.091 | 0.428 | 0.008 | 1.35 | 0.011 |

**Table 2.** Details of rainfall pattern, soil type, slope angle, anisotropy ratio, and layered slope of scenario analysis.

| Influential factors | Scenario values | | |
|---|---|---|---|
| Rainfall pattern | Constant rainfall intensity | | Variable rainfall intensity |
| Soil type | Soil 1 (sand) | Soil 2 (sandy loam) | Soil 3 (silt loam) |
| Slope angle, $\alpha$ | 5° | 10° | 20° |
| Anisotropy ratio* | 1 | 3 | 5 |
| Layered slope | Soil 1/Soil 3 | | Soil 3/Soil 1 |

*The anisotropy ratio is the ratio of horizontal hydraulic conductivity to vertical hydraulic conductivity and is set to a constant value.

There are two widely used definitions of upslope and downslope water flow: (1) defined in respect to the direction normal to the slope surface (Philip, 1991) (Fig. 3a); (2) defined with reference to the vertical direction (Jackson, 1992) (Fig. 3b). Since the second definition is considered to be more reasonable (Lu et al., 2011; Lv et al., 2013), the present study applied

the second definition to define the upslope and downslope flow (Fig. 3b).

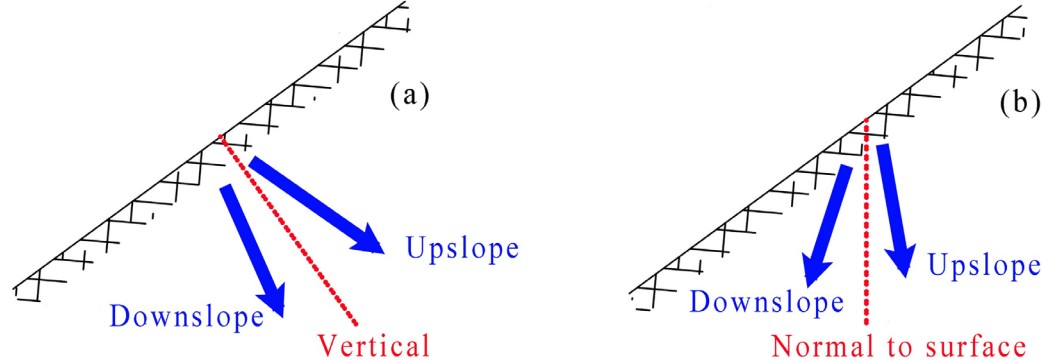

**Figure 3.** Definition of downslope and upslope flow directions. (a) defined in respect to the direction normal to the slope surface (Philip, 1991). (b) defined with reference to the vertical direction (Jackson, 1992). In this study, downslope and upslope flow is divided by the vertical direction.

**2.3 Modeling approach**

The present study used HYDRUS-2D/3D (Šimůnek et al., 2018) to simulate soil water infiltration on the slope. The HYDRUS-2D/3D model is effective in simulating the spatiotemporal distribution of soil water and can be used for various





ST (Nanda et al., 2018; Kesgin et al., 2020). The two-dimensional (2D) Richards equation can describe soil water flow in the slope domain induced by rainfall infiltration (Richards, 1931):

$$\frac{\partial \theta}{\partial t} = \frac{\partial}{\partial x_i} \left[ K(K_{ij}^A \frac{\partial h}{\partial x_j} + K_{iz}^A) \right] - S , \tag{1}$$

where $\theta$ is the volumetric water content [$L^3 L^{-3}$], $t$ is the time (T), $h$ is the pressure head [L] related to $\theta$, $K$ is the unsaturated hydraulic conductivity function [$L\ T^{-1}$], $x_i$ (i = 1, 2) represents coordinate directions (L), $K_{ij}^A$ represents components of a dimensionless anisotropy tensor $K^A$, and $S$ is a sink term accounting for root water uptake [$T^{-1}$] that was assumed to be zero in the present study. Eq. (1) represent water flow driven by the pressure head and gravity gradient.

The van Genuchten–Mualem model (Mualem, 1976; Van Genuchten, 1980) was used to describe the soil-water retention and hydraulic conductivity functions:

$$\theta(h) = \begin{cases} \theta_r + \frac{\theta_s - \theta_r}{[1+|\alpha h|^n]^m} & h < 0 \\ \theta_s & h \geq 0 \end{cases}, \tag{2}$$

$$K(h) = K_S S_e^l \left[ 1 - (1 - S_e^{\frac{1}{m}})^m \right]^2 , \tag{3}$$

$$S_e = \frac{\theta - \theta_r}{\theta_s - \theta_r} , \tag{4}$$

$$m = 1 - \frac{1}{n} \quad n > 1 , \tag{5}$$

in which $\theta_r$ and $\theta_s$ are the residual and saturated volumetric soil water contents [$L^3 L^{-3}$], respectively, $\alpha$ [$L^{-1}$], $n$, and $m$ are empirical parameters, $l$ is the pore-connectivity parameter with an average of ~0.5 for many soils (Liu et al., 2018; Zheng et al., 2021), $K_s$ [$L\ T^{-1}$] is the $K$ [$L\ T^{-1}$] value for the saturated condition, and $S_e$ is the effective saturation (-).

     The model domains of all scenario analyses had the same soil thickness (200 cm) and horizontal projection length (1,000 160   cm), whereas they had different slope lengths to ensure that the slopes of the different models received the same quantity of rainfall (Fig. 4). In particular, to ensure the observation of the response of deep soil layer water to rainfall within the simulated period of the layered slope system, the depth of the model domain was changed from the 200 cm to 100 cm, and two layers of soil with a thickness of 50 cm were used to represent the layered soil, with the layered interface parallel to the slope surface. The model domain for each slope was then established and discretized using triangles, with a finer element 165   mesh at the top to improve computational accuracy. The initial conditions were set as a hydrostatic equilibrium from the lowest nodal point (bottom pressure head value of –100 cm). Atmospheric flux boundary condition was applied at the top, and evaporation was ignored due to its small or insignificant effect among isolated rainfall events. No-flow boundaries were applied to both sides of the model domain, whereas a free drainage boundary was established at the bottom of the slope. The AR was set to a constant and the direction of the main axis of the anisotropy was aligned with the direction of the coordinate 170   axis in the model. Different RP, ST, SA, AR, LS were set in the model. Horizontal flux ($q_x$), vertical flux ($q_z$), and the deviation of the water flow direction from vertical (DWFFV) for the profile in the middle of the slope (A-A' in Fig. 4) were used to evaluate the effects of these factors on slope lateral water flow, and can be described as follows:



$$q_x = -K\frac{\partial \varphi}{\partial x}, \tag{6}$$

$$q_z = -K\frac{\partial \varphi}{\partial z}, \tag{7}$$

$$DWFFV = -\arctan\frac{q_x}{q_z}, \tag{8}$$

in which the variable $\varphi$ denotes soil water potential [L] defined by $\varphi = -h + z$, where z is elevation (positive upward).

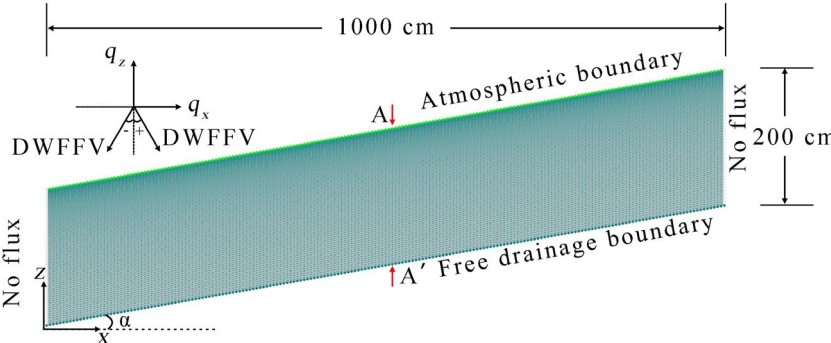

**Figure 4.** Physical domain, mesh, and boundary conditions. Positive $q_x$ and $q_z$ were set in the same direction as the coordinate axes. In the diagram, a positive and negative direction of water flow from the vertical (DWFFV) indicates
upslope and downslope, respectively.

## 3 Results

### 3.1 Dye tracer experiment

Figure 1c–e and f–h shows three cross-sectional photographs of the three points. Immediately after the cessation of the simulated rainfall, dyeing at Point 1 was mainly concentrated in the direction of the normal to slope surface and the vertical
direction of the injection point (Fig. 1c and Fig. 1f). However, the stained area shifted to the downslope direction one and two hours after the cessation of the simulated rainfall, with the furthest distance observed near the surface (Fig. 1d–e and Fig. 1g–h). This result indicated that soil water began to move downslope after the rainfall, with surface soil water first to initiate lateral downslope movement, gradually extending deeper over time. This phenomenon was also evident in the downslope movement of the stained area at the surface injection site (Fig. 1b).
On May 11, 2023, we excavated the embedded profiles at each depth to observe the patterns of soil water movement on the slope under multiple rainfall events. Due to wind erosion or dilution caused by multiple rainfall events, no traces of dye were observed at depths of 2 cm, 5 cm, 10 cm, and 20 cm. Traces were only evident at a depth of 50 cm, as shown in Fig. 2c (only displayed for the middle position of the slope; the dye staining pattern at the top and bottom positions was similar to that at the middle position). Through visual observation of the dye staining range, it was evident that under multiple rainfall events,
there is a pronounced lateral component in the soil water flow on the slope.



### 3.2 The effect of RP on the lateral water flow

The present study examined the spatial and temporal effects of RP on slope water infiltration under fixed ST, SA, and AR. Figure 5 provides a quantitative illustration (only showing SA = 5°, Soil 1, AR = 1; SA = 20°, Soil 2, AR = 1). At the start of rainfall, $q_x$ was positive and presented the upslope, with its maximum near the slope surface corresponding with a maximum upslope DWFFV. With continuing rainfall, the wetting front expanded towards the interior of the slope. The water flow behind the wetting front continued in an upslope direction and gradually approached a vertical direction near the slope surface. This was reflected in gradual decreases in $q_x$ and DWFFV near the slope surface and a slight increase in vertical fluxes. Under constant rainfall conditions (SA = 5°, Soil 1, and AR = 1), the wetting front moved to a depth of 6 cm at t = 10 min, and $q_x$, $q_z$, and the upslope DWFFV in the upper 6-cm layer varied between $3.45 \times 10^{-6}$ and $6.05 \times 10^{-4}$ cm min$^{-1}$ (Fig. 5a), $-7.21 \times 10^{-3}$ and $-1.13 \times 10^{-4}$ cm min$^{-1}$ (Fig. 5b), and 1.75 and 4.81° (Fig. 5c), respectively. At 420 min, the wetting front moved to a depth of 58 cm and $q_x$ decreased to $2.97 \times 10^{-4}$ cm min$^{-1}$ from $6.05 \times 10^{-4}$ cm min$^{-1}$ (10 min) near the surface (Fig. 5a). Correspondingly, $q_z$ at the slope surface increased to approximately $-8.27 \times 10^{-3}$ cm min$^{-1}$ (Fig. 5b). The upslope DWFFV decreased to about 2.05° (Fig. 5c). A decrease in rainfall intensity to 0 generally resulted in a change in the lateral water flow direction near the slope surface from upslope to downslope. This indicated a negative $q_x$ at the upper part of the profile, with the value gradually decreasing over time, whereas downslope DWFFV gradually increased. The downslope DWFFV at the surface varied from –53.73° to –60.94° from t = 600 min to t = 720 min (Fig. 5c). The downslope flow area extended deeper into the slope over time, and the region of changes in $q_x$, $q_z$, DWFFV coincide with the downward propagation of the wetting front over time. Under variable rainfall intensity (Fig. 5d–f and Fig. 5i–l), these patterns were consistent with the constant rainfall. The effects of ST and SA on the $q_x$, $q_z$, and DWFFV are discussed in Sections 3.3 and 3.4, respectively.

Figure 6 shows the changes in patterns of soil water content, water content change with time ($\partial\theta/\partial t$), $q_x$, $q_z$, and the horizontal gradient of soil water potential ($\partial\varphi/\partial x$) over time at different depths from the slope surface (5 cm, 10 cm, and 20 cm). The lateral upslope flux ($q_x$, $q_z$) near the slope surface exhibited a positive relationship with rainfall intensity during rainfall, subsequently diminishing as rainfall intensity decreased. Meanwhile, the direction of lateral water flow remained unchanged (Fig. 6a and Fig. 6b). An increase in depth resulted in a delay in the start of the increase or decrease in flux. After rainfall cessation, the direction of water flow at different depths gradually changed from lateral upward to lateral downward. Examining Fig. 6a as an illustration, the transition of $q_x$ from positive to negative after the cessation of rainfall indicates that downslope flow initiated almost immediately at a depth of 5 cm. At a depth of 10 cm, downslope flow initiation occurred at 516 min, and at a depth of 20 cm, it initiated at 717 min (Fig. 6a).





**Figure 5.** Simulated profiles of the wetting front in the middle of the slope at different times under constant rainfall intensity and variable rainfall intensity, considering fixed slope angle (SA), soil type (ST), and anisotropy ratios (AR): (a), (d), (g), (j) $q_x$ (positive values indicate upslope, negative values indicate downslope); (b), (e), (h), (k) $q_z$; (c), (f), (i), (l) deviation of the water flow direction from vertical (DWFFV) (positive values indicate upslope, negative values indicate downslope).





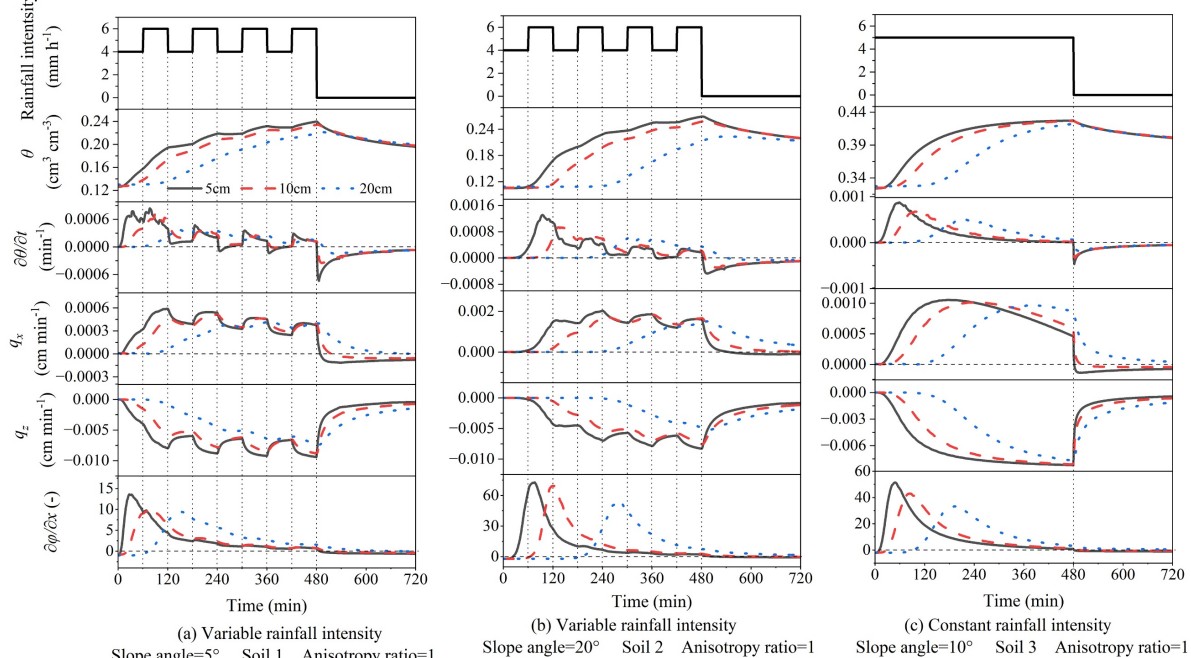

**Figure 6.** Simulations of soil water content ($\theta$), changes in water content over time ($\partial\theta/\partial t$), $q_x$, $q_z$, and change in soil water potential horizontal gradient ($\partial\varphi/\partial x$) over time for different depths within the middle section of the hillslope (A-A'). (a) variable rainfall, SA = 5°, Soil 1, and AR = 1; (b) variable rainfall, SA = 20°, Soil 2, AR = 1; (c) a constant rainfall, SA = 10°, Soil 3, and AR = 1.

### 3.3 The effect of ST on lateral water flow

Figure 7 shows the effect of ST on slope water flow by fixing RP, SA, and AR (i.e., constant rainfall intensity, SA = 10°, and AR = 1). Under a fixed ST, the region of changes in $q_x$, $q_z$, and DWFFV coincided with the downward propagation of the wetting front over time. However, ST affected the infiltration rate of the wetting front, with that under Soil 1 the fastest during which the depths of the wetting front reaching 18 cm, 30 cm, and 60 cm at 60 min, 120 min, and 420 min after the start of rainfall, respectively (Fig. 7a). In the wetting front reached 10 cm, 14 cm, and 34 cm for Soil 2 (Fig. 7d) and 14 cm, 22 cm, and 52 cm for Soil 3, respectively (Fig. 7g). Moreover, the maximum upslope $q_x$ at the slope surface during the simulation was $1.19 \times 10^{-3}$ cm min$^{-1}$ in Soil 3 (Fig. 7g), followed by those in Soil 1 ($1.12 \times 10^{-3}$ cm min$^{-1}$) (Fig. 7a) and Soil 2 ($1.11 \times 10^{-3}$ cm min$^{-1}$) (Fig. 7d). The maximum upslope DWFFV for Soil 1, Soil 2 and Soil 3 were 9.6°, 9.8 and 9.7°, respectively, indicating that the direction of water infiltration was almost normal to the slope surface.



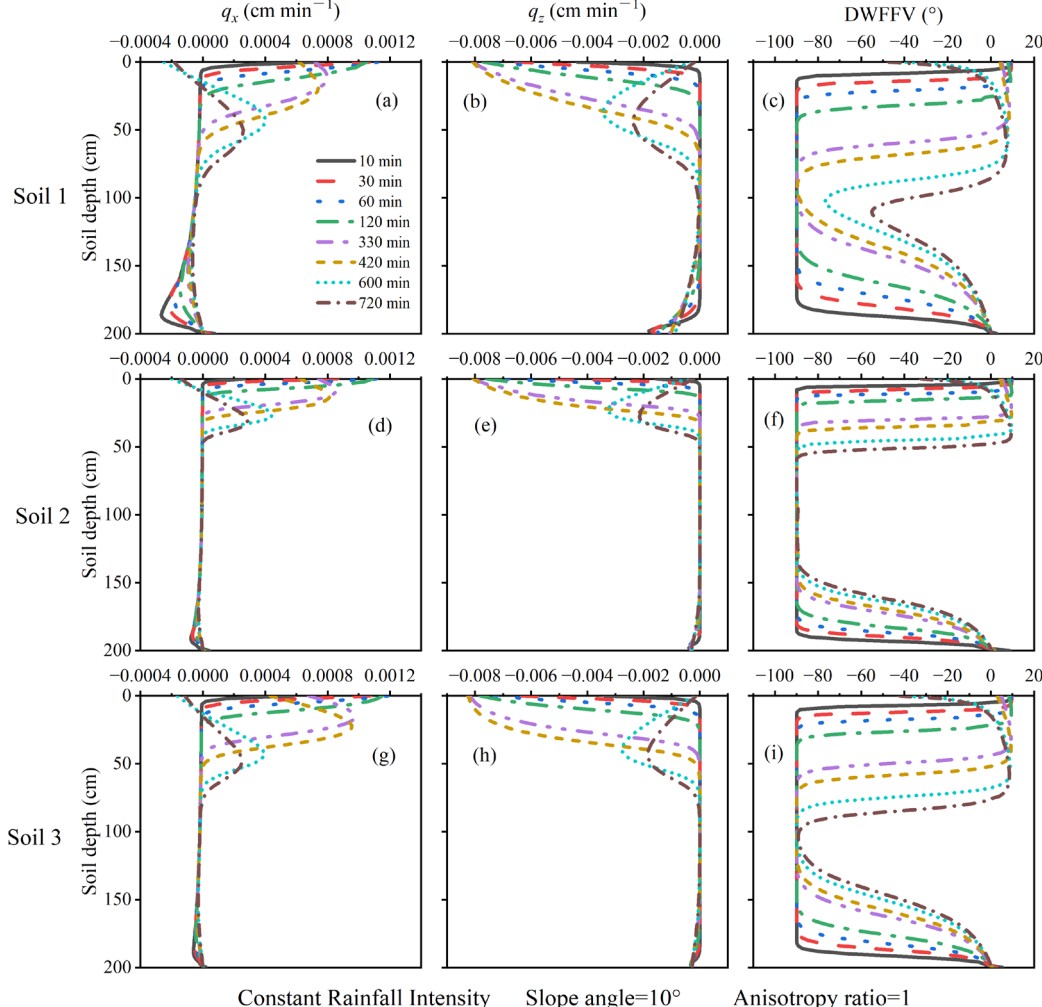

**Figure 7.** Simulated soil profiles distributions of the wetting front in the middle of the slope at different times under different soil type (ST) and fixed rainfall pattern(RP), slope angle (SA), and anisotropy ratios (AR):(a), (d), (g) $q_x$ (positive values indicate upslope, negative values indicate downslope), (b), (e), (h) $q_z$, (c), (f), (i) deviation of water flow direction from vertical (DWFFV) (positive values indicate upslope, negative values indicate downslope).

### 3.4 The effect of SA and AR on lateral water flow

Figure 8a–c shows the effects of SA on slope water flow under a fixed RP, ST, and AR (i.e., constant rainfall intensity, Soil 2, and AR = 1). $q_x$ and DWFFV represented the most substantial effects of SA on slope water flow. The values of $q_x$ and DWFFV showed positive correlations with SA over the entire simulation, consistent with the results of Lv et al. (2013). At t = 30 min, $q_x$ at the surface of the middle slope profile was $6.31 \times 10^{-4}$ cm min$^{-1}$ for a 5° slope, $1.65 \times 10^{-3}$ cm min$^{-1}$ for a 20°





slope (Fig. 8a), representing a 2.6-fold increase; DWFFV was 4.68° for a 5° slope and 19.28° for 20° slope (Fig. 8c),

representing a 4.12-fold increase. In comparison, spatial and temporal variations in $q_z$ were similar under different SA.

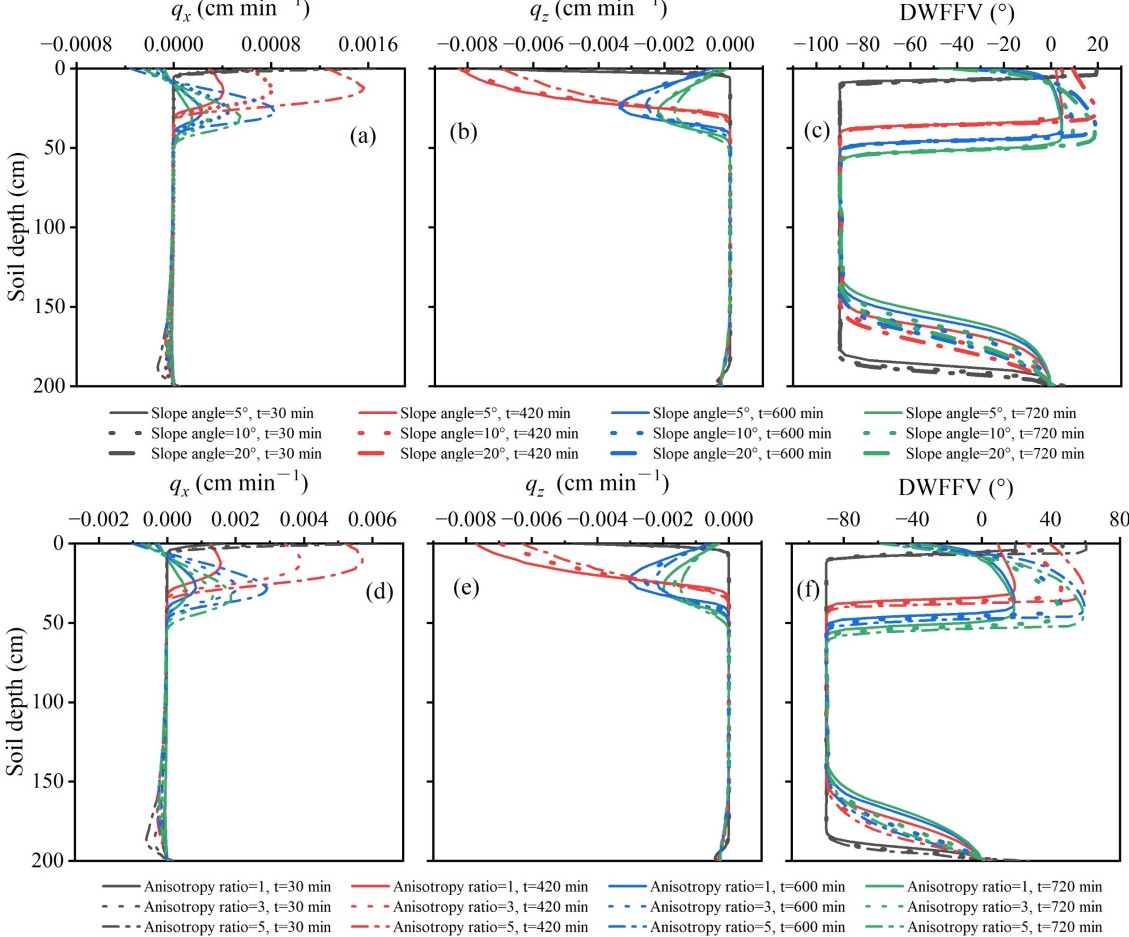

**Figure 8.** Simulated soil profile distributions in the middle of the slope at different times under different slope angle (SA) and fixed rainfall pattern (RP), soil type (ST), and anisotropy ratio (AR) (constant rainfall intensity, Soil 2, AR = 1) (a–c) and different AR under a fixed RP, ST, SA (constant rainfall intensity, SA = 20°, soil 2) (d–f). (a), (d), $q_x$ (positive values indicate upslope, negative values indicate downslope); (b), (e), $q_z$; (c), (f), water flow direction from vertical (DWFFV) (positive values indicate upslope, negative values indicate downslope).

The anisotropy of the soil medium is an additional important factor promoting lateral flow generation on slopes. Figure 8d–f shows the effect of AR on slope water flow under fixed RP, SA, and ST (i.e., constant rainfall intensity, SA = 20°, and Soil 2). AR had substantial impacts on the magnitude of $q_x$ and DWFFV. There was a positive correlation between the AR and the corresponding $q_x$. In contrast, the AR was negatively correlated with $q_z$. For instance, under an AR of 1 (i.e.,



isotropic conditions), the maximum $q_x$ and maximum $q_z$ at the slope surface were $2.12 \times 10^{-3}$ cm min$^{-1}$ (Fig. 8d) and $-7.7 \times$

$10^{-3}$ cm min$^{-1}$ (Fig. 8e), respectively. When the AR was increased to 5, the maximum $q_x$ increased to $7.55 \times 10^{-3}$ cm min$^{-1}$ (Fig. 8d), representing a 256% increase, whereas the maximum $q_z$ decreased to $-6.3 \times 10^{-3}$ cm min$^{-1}$ (Fig. 8e), indicating a 16.6% decrease compared to the values for AR = 1.

Under an AR = 1, the upslope DWFFV during the rainfall stage was smaller than the SA, and DWFFV increased with increasing AR (Fig. 8f). A change in AR from 3 to 5 increased the maximum upslope DWFFV from 46.8° to 60.6°, thereby

exceeding the SA. Similarly, after the cessation of rainfall, the downslope DWFFV increased with increasing AR (Fig. 8f). Although increased values of anisotropy does not alter the basic evolution of the iso-headline, it can amplify lateral flow by enhancing lateral conductivity (Mccord et al., 1991).

### 3.5 Water flow in unsaturated soil of layered slope systems

The slope water flow in layered system was examined by fixing the RP (constant rainfall conditions), SA (20°), and AR (1).

Figure 9 shows the spatial and temporal variations of $q_x$, $q_z$, DWFFV, pressure head, and soil water content for the Soil 1/Soil 3 (Fig. 9a) and Soil 3/Soil 1 (Fig. 9b) layered systems. Soil 1/Soil 3 and Soil 3/Soil 1 layered systems exhibit a linear pressure head at the initial moment, independent of soil type. However, water content is to be discontinuous at the layered interface, with soil water content increasing from 0.12 cm$^3$ cm$^{-3}$ in the overlying Soil 1 to 0.31 cm$^3$ cm$^{-3}$ in the underlying Soil 3 for the Soil 1/Soil 3 layered system. Conversely, the opposite occurred in the Soil 3/Soil 1 layered system. At the

initiation of rainfall, water infiltration on the slope of the layered system moved in the upslope direction, with this time coinciding with the largest $q_x$ and DWFFV values. With continuing rainfall, $q_x$ and DWFFV near the slope surface gradually decreased, and the wetting front gradually moved away from the slope surface. For the Soil 1/Soil 3 layered system, the wetting front reached the layered interface at 330 min; for the Soil 3/Soil 1 layered system, the wetting front reaches the interface at 420 min. After the cessation of rainfall, water flow near the slope surface gradually shifted downslope, similar to

the surface water flow patterns in single-layer soils observed earlier.





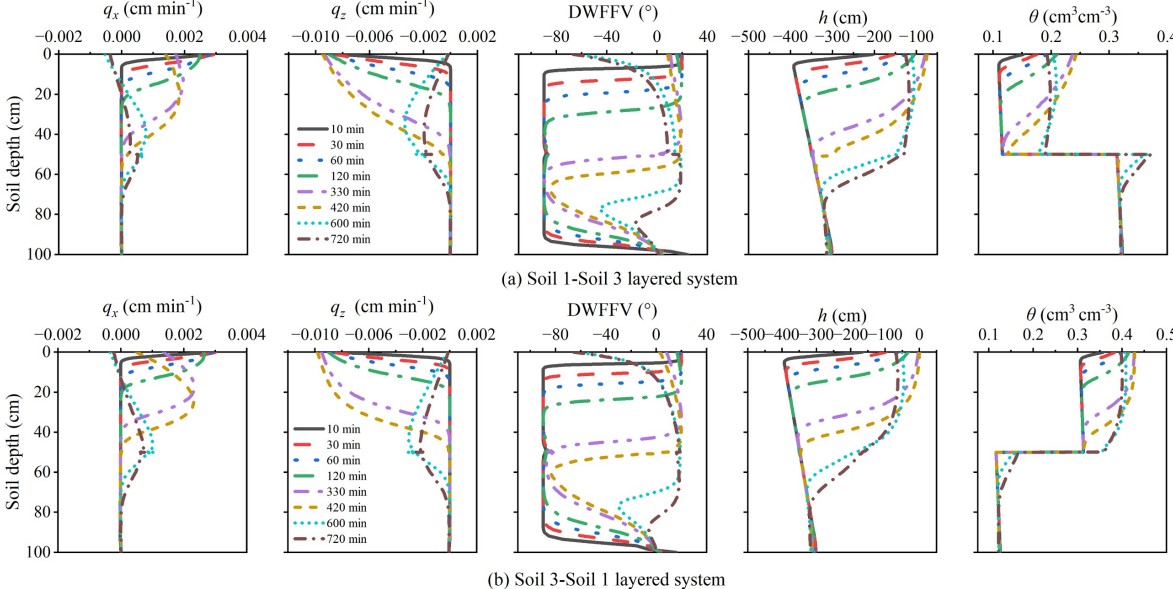

(a) Soil 1-Soil 3 layered system

(b) Soil 3-Soil 1 layered system

**Figure 9.** Simulated soil profile distributions in the middle of the slope at different times for the Soil 1/ Soil 3 (a), and Soil 3/Soil 1 (b) layered systems.

## 4 Discussion

A study by Lu et al. (2011) proposed a mechanism to determine the water flow direction on slopes based on the wetting, stable, and drying states in which the slope soils are located. They proposed that upslope lateral flow occurs when there is an increase in the water content of the soil at one point, near vertical flow occurs when the water content is quasi-steady, and lateral downslope flow occurs once the water content begins to decrease. Their observations were different from those of the present study. Taking the simulation of variable rainfall intensity under a SA of 5°, Soil 1, and AR of 1 as an example (Fig.

6a), rainfall intensity changed from 6 mm h$^{-1}$ to 4 mm h$^{-1}$ between 120 min and 180 min, 240 and 300 min, and 360 min and 420 min. $\partial\theta/\partial t$ at a depth of 5 cm was less than 0 between 243 min–265 min, and 362 min–409 min, respectively. Lateral water flow was in an upslope direction at these points rather than downslope. In addition, at the cessation of rainfall, soil water began to decrease at 481 min, 483 min, and 497 min at depths of 5 cm, 10 cm, and 20 cm, respectively. This indicated that from these moments onwards, $\partial\theta/\partial t$ for the corresponding depths dropped below 0. However, lateral downslope flows at

5 cm, 10 cm, and 20 cm began at 489 min, 516 min, and 717 min, respectively.

    The differences between the results of the current study and those of Lu et al. (2011) can be attributed to the methodology employed by Lu et al. (2011). In contrast to the present study, Lu et al. (2011) determined the direction of water flow at a given point based solely on the changes in soil water at that specific location and ignoring the changes in water content at adjacent points. The direction of slope water flow at any moment is determined by the instantaneous distribution of the soil





water content on the slope, independent of the state at the next moment. However, the conceptual illustration of flow regimes in a slope proposed by Lu et al. (2011) is very useful when determining the direction of water flow in a slope at a certain time (Fig. 10a–c). Figure 10 (d) and (e) show the distribution of soil water content under variable rainfall conditions for the mid-slope profile obtained by simulation at selected times (i.e., variable rainfall intensity, SA = 5°, Soil 1, AR = 1 for 250 min, 380 min, 720 min (Fig. 10d); variable rainfall intensity, SA = 20°, Soil 2, AR = 1 for 380 min, 720 min (Fig. 10e)).

$\partial\theta/\partial t$ near the slope surface was negative at these times. However, water near the slope surface flowed in the lateral upslope direction, except at 720 min. The soil water content behind the wetting front decreased with depth in the middle slope profile at 250 min and 380 min. This distribution of soil water content was similar to that shown in Fig.10b. The gradient of water content in the upslope direction between two points at the same elevation resulted in the lateral upslope flow. At 720 min, the distribution of soil water content behind the wetting front on the central slope was similar to that shown in Fig. 10c.

Therefore, water near the slope surface flowed in the downslope direction, whereas water in the deep soil layer flowed in the upslope direction.

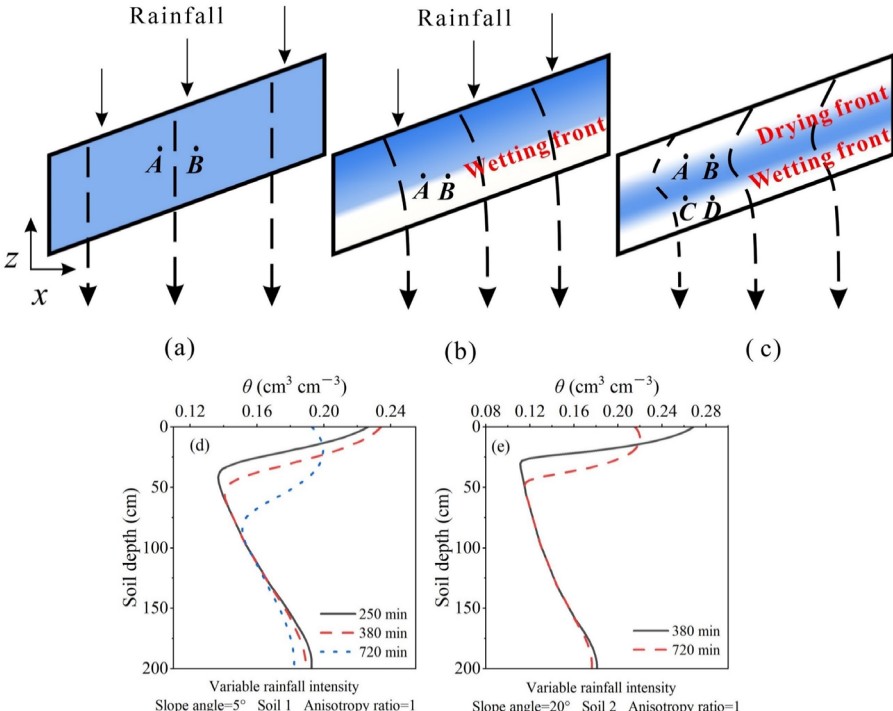

**Figure 10.** Conceptual illustration of flow regimes in a slope: (a) steady state, (b) wetting state, (c) drying and wetting state. Dashed lines with arrows show the hypothetical path of a particle in the flow field Lu et al. (2011). Points at the same

elevation (A and B; C and D) indicated that these points had the same gravitational potential. Distribution of soil water content in the middle profile of the slope at the selected times (d and e).





Lu et al. (2011) adopted a reasonable approach to determine the slope water flow direction based on the soil water gradient of two arbitrary points with the same elevation. This is because those two points have the same gravitational potential. Under variable saturation conditions, the effects of both gravitational and matric potential can result in complex patterns of water movement. Figure 11 shows the changes in soil water potential in the slope model domain (Fig. 11a: variable rainfall intensity, SA = 5°, Soil 1, and AR = 1; Fig. 11b: constant rainfall intensity, SA = 10°, Soil 3, and AR = 1; Fig. 11c: variable rainfall intensity, SA = 20°, Soil 2, AR = 1). At the onset of rainfall, the direction of soil water flow near the slope surface is oriented towards the lateral upslope direction (Fig. 11a-c at t = 30 min). As the rainfall continues, the wetting front gradually recedes from the slope surface. However, the iso-potential line near the slope surface gradually approached the horizontal direction, and the direction of water flow progressively changed to the vertical (Fig. 11a-c at t = 420 min). After the cessation of rainfall, the direction of water flow on the near-slope surface changed to downslope (Fig. 11a-c at t = 600 min and t = 720 min). This result was consistent with the changes in $q_x$, $q_z$, and DWFFV in the A-A' profile described in the previous sections (Fig. 5 and Fig. 7). During an isolated rainfall event, the soil water potential vertical gradient ($\partial\varphi/\partial z$) was negative when ignoring evaporation on the slope surface. Therefore, it was not possible to determine the direction of water flow on the slope based on $\partial\varphi/\partial z$. As shown in Fig. 11, upslope flow occurred when $\partial\varphi/\partial x$ was positive, whereas a zero and negative $\partial\varphi/\partial x$ resulted in water flow vertically downwards and towards the downslope, respectively. Fig. 6 also shows the change in $\partial\varphi/\partial x$ over time at different depths. There was a strong association between $q_x$ and soil water potential horizontal gradients over time. Under a variable rainfall, SA = 5°, Soil 1, and AR = 1, lateral upslope flow from the beginning of rainfall to 489 min at a depth of 5 cm corresponded with a positive $\partial\varphi/\partial x$, and lateral downslope flow from 489 min to 720 min corresponded to a negative value of $\partial\varphi/\partial x$ (Fig. 6a). Similarly, the other simulations showed an association between the direction of slope water movement and the direction of $\partial\varphi/\partial x$. $\partial\varphi/\partial x$ was positive when the direction of lateral water flow was upslope, and the direction of $\partial\varphi/\partial x$ changed in response to an alteration in the direction of infiltration from upslope to downslope. Thus, the direction of water lateral flow on the slope is determined by the direction of $\partial\varphi/\partial x$.

The above simulations analyses allow for the comprehensive explanation of three water flow patterns observed in the field, including upslope flow, vertical flow, and downslope flow during and after rainfall. At the beginning of rainfall, the direction of water flow near the slope surface was directed upslope. With continuing rainfall, the wetting front gradually moved away from the slope, resulting in the formation of a quasi-stable area behind the wetting front. In this zone, the direction of water flow gradually changes to the vertical. At the cessation of rainfall, the direction of water flow on the slope gradually changed to the lateral downslope. Thus, at the end of the rainfall event, the staining at point 1 was mainly concentrated in the normal and vertical directions (Fig. 1c and Fig. 1f), and the stained area shifted to the lateral downslope, with the furthest distance observed near the surface, 1 and 2 h after the cessation of rainfall, respectively (Fig. 1d–e and Fig. 1g–h). Furthermore, the modeling of water flow in slopes under different conditions confirmed the existence of three distinct water flow patterns that manifested during rainfall.





(a) Variable rainfall intensity

Slop angle=5° Soil 1 Anisotropy ration=1

(b) Conatant rainfall intensity

Slop angle=10° Soil 3 Anisotropy ration=1

(c) Variable rainfall intensity

Slop angle=20° Soil 2 Anisotropy ration=1





**Figure 11.** Progression of the distribution of $\varphi$ (soil water potential) (units: cm) over time for the slope model domain. The blue arrow represents the direction of the soil water potential gradient, i.e., the direction of water flow. (a) variable rainfall, SA = 5°, Soil 1, and AR = 1; (b) constant rainfall, SA = 10°, Soil 3, and AR = 1; (c) variable rainfall, SA = 20°, Soil 2, and AR = 1.

**5 Conclusions**

This study conducted a rainfall tracer experiment on the slope to investigate water flow patterns. The results revealed the presence of three water flow patterns within the slope, namely upslope and vertical flow during rainfall and downslope flow after cessation of rainfall. At the beginning of rainfall, the direction of water flow near the slope surface was towards the upslope. As rainfall continued, the wetting front gradually moved away from the slope surface, leading to the formation of a quasi-stable zone behind the wetting front. In this zone, the direction of water flow progressively shifted towards the vertical.

At the cessation of rainfall, the direction of water flow on the slope surface gradually changed to the downslope. Furthermore, the simulation of water flow on slopes under different conditions confirmed the existence of three distinct water flow patterns that manifest within slopes during rainfall. The model simulations also demonstrated that a decrease in rainfall intensity resulted in decreases in $q_x$ and $q_z$ near the slope surface. In addition, different soil types showed different unsaturated hydraulic conductivities, resulting in differences in the water infiltration rate. Among the three soil types, Soil 1

showed the fastest wetting front propagation rate, whereas Soil 3 showed a larger $q_x$ throughout the simulation. $q_x$ and DWFFV represented the most substantial effects of SA and AR on slope water flow. Over the entire simulation period, DWFFV increased with SA and AR, resulting in a simultaneous increase in the corresponding $q_x$. For layered slopes in initial hydrostatic equilibrium, the changes in soil water flow were similar to those in single-layer soils. The results of this study showed that the boundary conditions and the wetting, stable, and drying states were not drivers of the synchronous change in

the lateral water flow direction. The direction of lateral water flow exhibited a complete consistency with the horizontal gradient of soil water potential. Therefore, the direction of lateral water flow could be determined by the $\partial\varphi/\partial x$.

*Data availability*

All the data needed to evaluate the conclusions in the paper are present in the paper and are contained in Figs.5–11.

*Author contributions*

WG: Conceptualization, Data curation, Software, Methodology, Writing - original draft. YM, YL, CZ and CM: Writing - review& editing. Supervision. XL: Conceptualization, Writing - review & editing, Supervision, Funding acquisition. YC: Data curation.





## Acknowledgments

This study was financially supported by the Fundamental Research Funds for the Central Universities, China [grant
number 300102292904], the China Postdoctoral Science Foundation [grant number 2022M720536]. The authors appreciate
the editors and anonymous reviewers for their valuable comments and suggestions that helped improve the manuscript.

## Competing interests

The authors declare that they have no known competing financial interests or personal relationships that could have appeared
to influence the work reported in this paper.

## Appendix. List of Abbreviation:

| | |
|---|---|
| RP | Rainfall patterns |
| ST | Soil types |
| SA | Slope angle |
| AR | Anisotropy ratio |
| LS | Layered slope |
| $qx$ | Horizontal flux |
| $qz$ | Vertical flux |
| DWFFV | Deviation of the water flow direction from vertical |
| $\partial\varphi/\partial x$ | Soil water potential horizontal gradient |
| $\partial\varphi/\partial z$ | Soil water potential vertical gradient |
| $\partial\theta/\partial t$ | Rate of water content change as a function of time |

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
