# Peer review of "Comprehensive Factors Influencing Lateral Soil Water Flow Patterns on Hillslopes: Insights from Experimental and Simulation Studies"

_Hydrology and Earth System Sciences, 2024_

## Author Comment (AC1)

**Point-by-point Responses for Reviewer 1 for hess-2024-19**

This study implemented a rainfall tracer experiment on a slope to analyze water flow patterns. Simultaneously, the HYDRUS-2D model was employed for scenario simulations, encompassing two rainfall patterns (RP), three soil types (ST), three slope angles (SA), three anisotropy ratios (AR), and two layered slope systems (LS), aiming to deepen comprehension of the those factors affecting lateral water flow on slopes. It has a potential contribution to the community as this topic is not well covered in the previous research, e.g., simulating of subsurface lateral flow based on the observation data. However, the current paper has some deficiencies:

Response: We thank the referee of reading the manuscript and providing helpful comments and suggestions. Here we reply to the comments point-by-point. We hope that these changes satisfy the requirements for proceeding with the publication of the updated manuscript.

I. A significant concern arises from the lack of connection between the experimental data and the model's approach or outputs. Essentially, the model remains unvalidated or untested against experimental data. Furthermore, even for tracer experiments, there was no data provided apart from a picture.

Response: Thank you for your valuable feedback. We acknowledge the concern regarding the lack of connection between the experimental data and the model's approach or outputs. Field experiments primarily serve for qualitative observations of slope soil moisture behavior. They reveal that during the early stages of rainfall, moisture moves upslope, gradually stabilizes, then transitions vertically downward. After rainfall ceases, it flows downslope. However, it's important to emphasize that these observations were made under specific conditions in the field experiments.

In contrast, our two-dimensional numerical model offers a powerful tool for analyzing soil moisture movement under various conditions. To investigate whether similar soil moisture behavior occurs under different circumstances, we conducted scenario simulations. The results from our modeling work provided insights into the changes in the direction of lateral water flow during rainfall under varying conditions.

This approach allowed us to combine qualitative field observations with quantitative modeling to gain a more comprehensive understanding of slope soil moisture dynamics during rainfall events. We believe this combination strengthens the robustness and reliability of our study findings.

Additionally, we established an in-situ monitoring station at the Hongshixia Botanical Garden in Mu Us Sandy Land, China. Soil moisture automatic monitoring probes were installed at six monitoring points on both the windward and leeward slopes of sand dunes, covering the top, middle, and down of the slope. At each location, six soil moisture probes were horizontally installed at depths of 10, 20, 50, 80, 100, and 150 cm, as illustrated in Figure 1.

We employed Hydrus-2D to establish a hydraulic model for simulating soil moisture dynamics on sand dune slopes. The model was calibrated and validated using measured soil moisture data. We evaluated the fitting performance of the model using the root mean square error ($RMSE$) and the agreement of index ($d$), as shown in Figures 2 and 3, respectively.

[Figure]

Figure 1. Schematic of the in-situ monitoring station. WW-up, windward up-slope; WW-middle, windward middle-slope; WW-down, windward down-slope; LW-up, leeward up-slope; LW-middle, leeward middle-slope; and LW-down, leeward down-slope.

[Figure]

Figure 2. Simulated and measured soil moisture during the calibration period

[Figure]

Figure 3. Simulated and measured soil moisture during the validation period

We selected a typical rainfall event to analyze the variation of soil water potential on the slope over time during the rainfall process, aiming to determine changes in soil water flow direction. Additionally, we

employed flow particles to visually observe the flow trajectory of soil moisture on the slope, as depicted in Figure 4. At t=0, 1 h before the onset of rainfall, and at t=2, 5, and 9h, during the rainfall process. Furthermore, at t=30, 54, 78, and 102h, representing moments after rainfall cessation. Through analysis of soil water potential changes and flow particle trajectories, we identified three patterns of soil water movement on the slope: lateral upslope and vertical flow during rainfall, followed by lateral downslope flow after rainfall.

The detailed information of these research results will be provided in the next study.

[Figure]

Figure 4. Soil water potential in the model domain during the rainfall event (vertical scaling factor is 1.8). The blue arrows indicate the gradient direction of soil water potential, representing the direction of soil water flow. The red dots and red lines depict tracer particles and their trajectories, respectively.

II. The paper contains an excessive amount of abbreviations, leading to poor readability. The introduction lacks significance, and the contents is not good enough to clearly specify the research's contributions. Furthermore, the literature review is outdated and fails to reflect the recent progress in related research.

Response: Thank you for your feedback. We appreciate your comments regarding the excessive use of abbreviations, readability issues, and the need for a clearer presentation of the research contributions, significance, and updated literature review. In response, we will address these concerns in the revised manuscript.

We will provide a comprehensive list of abbreviations at the end of the manuscript to improve readability and reduce confusion, as follows:

Appendix. List of Abbreviation:

| | |
|---|---|
| RP | Rainfall patterns |
| ST | Soil types |
| SA | Slope angles |
| AR | Anisotropy ratios |
| LS | Layered slope |
| $q_x$, | Horizontal flux |
| $q_z$ | Vertical flux |
| DWFFV | Deviation of the water flow direction from vertical |
| $\partial\varphi/\partial x$ | Soil water potential horizontal gradient |
| $\partial\varphi/\partial z$ | Soil water potential vertical gradient |
| $\partial\theta/\partial t$ | Rate of moisture content change as a function of time |

III. The term "deviation of water flow direction from vertical (DWFFV)" could potentially be effectively assessed in HYDRUS by considering the velocity vector. The concept of a layered slope system appears challenging to interpret or assess. It appears to be described as altering the depth of the model domain from 200 cm to 100 cm and incorporating two layers of soil, each 50 cm thick, to represent layered soil, with the interfaces between layers aligned parallel to the slope surface. However, this description lacks clarity for me.

Response: Thank you for your feedback. Indeed, the velocity vector in HYDRUS represents the direction of soil water flow. In this study, we use the vertical direction as the boundary between upslope and downslope lateral flow, with positive DWFFV values indicating lateral upslope flow and negative values representing lateral downslope flow. The magnitude of DWFFV denotes the degree of deviation from the vertical direction. Additionally, we utilize horizontal flux, vertical flux, and the angle of deviation of water flow direction from vertical (DWFFV) to quantitatively analyze the impacts of rainfall patterns, soil types, slope angles, and anisotropy ratios on lateral soil moisture infiltration on the slope.

As for the layered slope system, we understand the necessity for clarity in our description. To observe soil moisture flow at layer interfaces, we reduced the model domain depth from 200 cm to 100 cm, incorporating two soil layers, each 50 cm thick. We aim to illustrate this layered slope system in Figure 5 and will provide clearer explanations, possibly supplemented with visual aids, in the revised manuscript to enhance comprehension.

[Figure]

Figure 5. Physical domain, mesh, and boundary conditions for layered slope system.

IV. The finding lacks novelty. I observed that the outcomes demonstrated "complete consistency between the direction of lateral water flow (lateral unsaturated upslope or downslope flow) and soil water potential horizontal gradients ($\partial \varphi / \partial x$)." Nevertheless, what sets this finding apart as novel?

Response: Thank you for your valuable comments. In the revised manuscript, we will provide a more explicit explanation of the main conclusion regarding lateral water flow on homogeneous isotropic slope. Specifically, we will emphasize that the movement of soil moisture in slope is driven by both gravitational potential and matrix potential. The inconsistency in the directions of gravitational potential gradient and matrix potential gradient leads to lateral flow of soil water on the slope with the angle of deviation from the vertical direction determined by the competition between these gradients. Additionally, we will clarify that changes in boundary conditions and the transition between wetting, stable, and drying states are not synchronous drivers of changes in the direction of lateral soil moisture flow on the slope. At any point on the slope surface, the horizontal gradient of water potential determines the direction of lateral soil moisture flow.

V. To sum up, the paper lacks a clear purpose. I'm unsure about the problem it aimed to address.

Response: Thank you for your feedback. In unsaturated slopes, soil water flow is driven by gravitational and matrix potentials, with the gradients of these potentials determining the direction of soil water flow. Typically, the gradient of the matrix potential does not align with the vertical direction, leading to soil water flow deviating from the vertical direction. Under transient rainfall conditions, unsaturated slopes exhibit complex spatiotemporal patterns of soil water flow, where lateral upslope flow, vertical flow, and lateral downslope flow may coexist at different locations on the slope simultaneously. However, there are still some deficiencies in understanding the lateral soil water flow processes on slope.

Moreover, most previous studies have primarily focused on vertical infiltration of soil moisture on slope, overlooking the lateral flow processes. Therefore, the objectives of this study are twofold: (1) to analyze the lateral flow processes of soil moisture on slope using rainfall dye tracer experiments combined with scenario simulations, and (2) to investigate the effect of different rainfall patterns (RP), slope angles (SA), soil types (ST), anisotropy ratios (AR), and layered slope systems (LS) on slope lateral water flow. Additionally, we will address these concerns in the revised manuscript by providing supplementary explanations and modifications to ensure clarity regarding the purpose and objectives of the study.

More details:

1. Fig 5., the legend did not indicate the difference of slope angle.

Response: Thank you for your feedback. In this section, we primarily focus on analyzing the impact of rainfall patterns (constant and variable rainfall intensity) on the lateral infiltration of soil moisture on the slope. Therefore, we kept other influencing factors such as slope angle, soil type, and anisotropy ratio constant. The influence of slope angle on lateral infiltration of soil moisture is discussed in detail in Section 3.4 of the manuscript.

2. Fig 11., Anisotropy ration should be anisotropy ratio.

Response: sorry, we will correct it.

3. It is unclear why "At the beginning of rainfall, the direction of water flow near the slope surface was

towards the upslope."

Response: Under transient rainfall conditions, soil moisture flow is driven by both gravitational potential and matric potential. In rainfall event, for homogeneous isotropic flat soil, the total potential gradient driving soil moisture movement is typically vertical, resulting in vertical upward or downward flow with no significant lateral flow. However, in homogeneous isotropic slopes, the gravitational potential gradient remains constant (-1) at any position along the slope, oriented vertically. In contrast, the matric potential gradient, closely related to soil moisture content, can vary significantly and is influenced by the wetting and drying processes of the slope, leading to the direction of slope water movement lying within the angular range between the matric potential gradient and the gravitational potential gradient. At the onset of rainfall, a matric potential gradient directed inward toward the normal slope surface is established, causing the combined force of gravity and matric suction to act in the upslope direction near the slope surface, hence the direction of water flow near the slope surface towards the upslope. As the rainfall continues, the soil moisture flow near the slope gradually transitions to a quasi-steady state, where the moisture flow near the slope is primarily driven by gravity in a quasi-vertical direction. After the rainfall ends, a gradient of matric potential forms in the downslope direction near the slope, leading to lateral downslope water flow.

We will supplement the explanation in the revised manuscript.

---

## Author Comment (AC2)

**Point-by-point Responses for Reviewer 2 for hess-2024-19**

This paper investigates the factors influencing lateral soil water flow on hillslope by both experimental and simulation methods. It is an interesting topic and is widely concerned. However, there are several problems that needs to be revised.

We appreciate your detailed comments and suggestions. All recommended corrections and modifications have been implemented. We followed the guidelines to craft this response and furnished a point-by-point reply to your comments. All specific modifications have been made in the revised manuscript.

1. The main conclusion "The direction of lateral water flow is regulated by $\delta\psi/\delta x$ rather than the change in water content over time on hillslope" is easy to obtain according to basic soil physics and is not new to me. I suggest the authors to express this more explicit.

Response: Thank you for your suggestion. In the revised manuscript, we will provide a more explicit explanation of the main conclusion regarding lateral water flow on homogeneous isotropic slope. Specifically, we will emphasize that the movement of soil moisture in slope is driven by both gravitational potential and matrix potential. The inconsistency in the directions of gravitational potential gradient and matrix potential gradient leads to lateral flow of soil moisture on the slope with the angle of deviation from the vertical direction determined by the competition between these gradients. Additionally, we will clarify that changes in boundary conditions and the transition between wetting, stable, and drying states are not synchronous drivers of changes in the direction of lateral soil moisture flow on the slope. At any point on the slope surface, the horizontal gradient of water potential determines the direction of lateral soil moisture flow.

2. Lines 27-28. The factors influencing lateral flow could be more specific.

Response: Thank you for your valuable comments. The infiltration of soil moisture on slope is a complex and dynamic process influenced by various factors, including soil properties (such as texture, bulk density, and hydraulic conductivity), rainfall characteristics (including intensity and duration), terrain features (such as slope angle, aspect, and surface roughness), initial soil moisture conditions, vegetation cover, land use, and more. Furthermore, most previous studies have predominantly concentrated on the vertical infiltration of soil moisture on slope surfaces, neglecting the lateral flow processes of soil moisture on the slope. This study primarily investigates the impacts of rainfall patterns, slope gradient, soil types, anisotropy ratio, and layered slope systems on the lateral movement of soil moisture on slope. We will provide more specific details on the factors influencing lateral flow in the revised manuscript.

3. In the manuscript, both field experimental and numerical simulation are conducted to study the factors influencing lateral soil water flow patterns. What's the relationship between the filed experimental and numerical simulation is not clear. The manuscript seems mainly focus on the numerical simulation.

Response: Thank you for your valuable comments. Field experiments primarily serve for qualitative observations of slope soil moisture behavior. They reveal that during the early stages of rainfall, moisture moves upslope, gradually stabilizes, then transitions vertically downward. After rainfall ceases, it flows downslope. However, it's important to emphasize that these observations were made under specific

conditions in the field experiments.

In contrast, our two-dimensional numerical model offers a powerful tool for analyzing soil moisture movement under various conditions. To investigate whether similar soil moisture behavior occurs under different circumstances, we conducted scenario simulations. The results from our modeling work provided insights into the changes in the direction of lateral water flow during rainfall under varying conditions.

This approach allowed us to combine qualitative field observations with quantitative modeling to gain a more comprehensive understanding of slope soil moisture dynamics during rainfall events. We believe this combination strengthens the robustness and reliability of our study findings.

Additionally, we established an in-situ monitoring station at the Hongshixia Botanical Garden in in Mu Us Sandy Land, China. Soil moisture automatic monitoring probes were installed at six monitoring points on both the windward and leeward slopes of sand dunes, covering the top, middle, and down of the slope. At each location, six soil moisture probes were horizontally installed at depths of 10, 20, 50, 80, 100, and 150 cm, as illustrated in Figure 1.

[Figure]

Figure 1. Schematic of the in-situ monitoring station. WW-up, windward up-slope; WW-middle, windward middle-slope; WW-down, windward down-slope; LW-up, leeward up-slope; LW-middle, leeward middle-slope; and LW-down, leeward down-slope.

We employed Hydrus-2D to establish a hydraulic model for simulating soil moisture dynamics on sand dune slopes. The model was calibrated and validated using measured soil moisture data. We evaluated the fitting performance of the model using the root mean square error (*RMSE*) and the agreement of index (*d*), as shown in Figures 2 and 3, respectively.

[Figure]

Figure 2. Simulated and measured soil moisture during the calibration period

[Figure]

Figure 3. Simulated and measured soil moisture during the validation period

We selected a typical rainfall event to analyze the variation of soil water potential on the slope over time during the rainfall process, aiming to determine changes in soil water flow direction. Additionally, we employed flow particles to visually observe the flow trajectory of soil moisture on the slope, as depicted in Figure 4. At t=0, 1 h before the onset of rainfall, and at t=2, 5, and 9h, during the rainfall process. Furthermore, at t=30, 54, 78, and 102h, representing moments after rainfall cessation. Through analysis of soil water potential changes and flow particle trajectories, we identified three patterns of soil water movement on the slope: lateral upslope and vertical flow during rainfall, followed by lateral downslope flow after rainfall.

The detailed information of these research results will be provided in the next study.

[Figure]

Figure 4. Soil water potential in the model domain during the rainfall event (vertical scaling factor 1.8). The blue arrows indicate the gradient direction of soil water potential, representing the direction of soil water flow. The red dots and red lines depict tracer particles and their trajectories, respectively.

4. Lines 85-100. The rainfall dye tracer was conducted with much heavy rainfall with an average rate of 0.3125 cm/min which is about 50 times the value for numerical simulation. The rainfall is too large to the real rainfall and may make saturated flow. Meanwhile, the soil water content is needed to show the unsaturated soil water condition during the experiment.

Besides, the soil texture and hydraulic properties should also be presented.

Response: Thank you for your valuable comments and suggestions. Yes, the rainfall intensity (0.3125 cm/min) used in the rainfall dye tracer experiment is indeed much higher than what would typically occur in natural rainfall. In our scenario simulations, we considered two rainfall modes: constant rainfall intensity and variable rainfall intensity. Saturated hydraulic conductivities for the three soil types were as follows: 0.783 cm min$^{-1}$ (equivalent to 459.8 mm h$^{-1}$) for Sand, 0.155 cm min$^{-1}$ (93 mm h$^{-1}$) for Sandy loam, and 0.011 cm min$^{-1}$ (6.6 mm h$^{-1}$) for Silt loam. The maximum rainfall intensity was set at 6 mm h$^{-1}$, which was below the saturated hydraulic conductivities of these three soil types, ensuring that the flow system remained consistently unsaturated. As written in the manuscript: "No runoff from the slope occurred during the rainfall process."

In this study, we utilized sand (soil 1) with the same characteristics as the sand dunes where the dye tracer experiments were conducted. The rainfall intensity for the dye tracer experiment needed to be

lower than the saturated hydraulic conductivity of the sand (soil 1), theoretically preventing saturated flow on the slope. Furthermore, we did not observe any occurrence of saturated flow during the dye tracer experiment.

We used the characteristics of sand (soil 1) from the adjacent area study to represent the soil properties of the sand dune slope where the dye tracer experiments were conducted. In the revised manuscript, we will provide the soil water retention curve and hydraulic conductivity function for the three soil types.

5. Fig 1. It's better to mark the direction of slope foot in fig (a) and (b).

Response: Thank you for the suggestion. We will mark the direction of the slope foot in Fig 1(a) and (b) in the revised manuscript.

6. Fig 3 (a) is normal to surface, (b) is vertical?

Response: We apologize for the oversight. Indeed, Fig 3 (a) is normal to surface, (b) is vertical. We will make the necessary corrections.

7. In line 175 DWFFV=-arctan($q_x/q_z$), when $q_z$<0 and $q_x$>0, DWFFV>0. The signs seem reversed in Fig 5, where positive $q_x$ and negative $q_z$ corresponds to negative DWFFV.

Response: Thank you for your feedback. Indeed, DWFFV is greater than 0 when $q_z$ < 0 and $q_x$ > 0. Initially, all models assume hydrostatic equilibrium from the lowest nodal point (bottom pressure head value of –100 cm), resulting in DWFFV being at –90°. It's noteworthy that the bottom boundary of the model domain features a free drainage condition (with matrix potential gradient of 0), which influences the DWFFV value at the bottom of the model. While DWFFV is approximately –90°, the values of $q_z$ and $q_x$ are very small, both being less than 0. As rainfall begins, water flow at the slope's top shifts laterally upslope, corresponding to positive $q_x$ and negative $q_z$ values. Meanwhile, in deeper layers unaffected by rainfall, DWFFV approximates –90°, corresponding to negative $q_x$ and $q_z$ values, albeit very small in magnitude. Over time, the lateral upslope flow region gradually extends deeper.

8. Lines 200. What is duration of the rainfall, when rainfall ceases.

Response: Thank you for pointing that out. The duration of the rainfall and the simulation period have been clarified in the revised manuscript. The two rainfall patterns investigated were as follows:

(1) Rainfall intensity in the first 8 hours was equal to 0.5 cm h$^{-1}$ (constant rainfall intensity).

(2) Rainfall intensity varied between 0.4 cm h$^{-1}$ and 0.6 cm h$^{-1}$ over an 8-hour period, changing hourly while remaining constant within each hour (variable rainfall intensity).

Both rainfall patterns had a total duration of 8 hours. The simulation extended from the start of rainfall to 4 hours after the rainfall ceased, resulting in a total simulation period of 12 hours.

9. Fig 10. What does the blue color mean? The sub-figure number (d-e) should be consistent with (a-c).

Response: Thank you for your question. Consider two adjacent points A and B in a homogeneous and isotropic hillslope with the same elevation near the surface of the hillslope. If the two points reach a

steady state (i.e., the moisture content becomes invariant with time) and assuming the rainfall intensity is less than the saturated hydraulic conductivity of the slope materials, then the water flux at these two points should be equal to the constant vertical infiltration rate at the surface, leading to no gradient in matric potential or moisture content between the two points. Thus, the resulting unsaturated flow is predominantly vertical because of gravity, as shown in Figure 10a. If the rainfall intensity increases to another value (still less than the saturated hydraulic conductivity) at some time $t_0$, soil at the surface becomes wetter leading to an additional gradient of moisture content in the direction normal to and inward from the slope surface. This will result immediately in upslope lateral flow in the soil adjacent to the surface but not in the region of points A and B. Flow at points A and B will remain vertical under gravity at time $t_0$. In time, the region of upslope lateral flow will propagate into the slope directly normal and inward from the surface as shown in Figure 10b. The additional wetting front will arrive earlier at point A than at point B as the slope normal distance from the surface to point A is shorter than that to point B, yielding a gradient in matric potential between points A and B. When this happens, upslope lateral flow occurs. Using the same logic, if the rainfall intensity decreases or the rainfall ceases, the opposite will occur as point A will drain earlier than point B, leading to downslope lateral flow as shown in Figure 10c. At the same time, on the slope-parallel horizon between points C and D shown in Figure 10c, the wetting front has just arrived, leading to lateral upslope flow by the same mechanism shown in Figure 10b under the wetting state.

Additionally, we will ensure that the sub-figure numbering (d-e) is consistent with (a-c). Necessary adjustments will be made in the revised manuscript.

10. Line 360. (b) "Conatant" should be "Constant".

Response: sorry, we will correct it.

Lu, N., Kaya, B.S., Godt, J.W., 2011. Direction of unsaturated flow in a homogeneous and isotropic hillslope. Water Resour. Res. 47(2): 1205-1209. https://doi.org/10.1029/2010wr010003.